# Impact of Structural Alterations from Chemical Doping on the Electrical Transport Properties of Conjugated Polymers

**DOI:** 10.3390/polym16172467

**Published:** 2024-08-30

**Authors:** Baiqiao Yue, Xiaoxuan Zhang, Kaiqing Lu, Haibao Ma, Chen Chen, Yue Lin

**Affiliations:** 1College of Chemistry, Fuzhou University, Fuzhou 350002, China; 211327018@fzu.edu.cn (B.Y.);; 2CAS Key Laboratory of Design and Assembly of Functional Nanostructures, Fujian Key Laboratory of Nanomaterials, and State Key Laboratory of Structural Chemistry, Fujian Institute of Research on the Structure of Matter, Chinese Academy of Sciences, Fuzhou 350002, China; 3Fujian College, University of Chinese Academy of Sciences, Fuzhou 350002, China; 4College of Chemistry and Materials Science, Fujian Normal University, Fuzhou 350007, China; 5Science and Technology on Advanced Ceramic Fibers and Composites Laboratory, College of Aerospace Science and Engineering, National University of Defense Technology, Changsha 410073, China; 6Fujian Science & Technology Innovation Laboratory for Optoelectronic Information of China, Fuzhou 350108, China

**Keywords:** conjugated polymers, thermoelectric materials, carrier transport, backbone planarity and stiffness

## Abstract

Conjugated polymers (CPs) are widely used as conductive materials in various applications, with their conductive properties adjustable through chemical doping. While doping enhances the thermoelectric properties of CPs due to improved main-chain transport, overdoping can distort the polymer structure, increasing energy disorder and impeding intrinsic electrical transport. This study explored how different dopants affect the structural integrity and electrical transport properties of CPs. We found that dopants vary in their impact on CP structure, consequently altering their electrical transport capabilities. Specifically, ferric chloride (FeCl_3_)-doped indacenodithiophene-*co*-benzothiadiazole (IDTBT) shows superior electrical transport properties to triethyloxonium hexachloroantimonate (OA)-doped IDTBT due to enhanced backbone planarity and rigidity, which facilitate carrier transport and lower energetic disorder. These results highlight the critical role of dopant selection in optimizing CPs for advanced applications, suggesting that strategic dopant choices can significantly refine the charge transport characteristics of CPs, paving the way for their industrialization.

## 1. Introduction

Conjugated polymers (CPs) have emerged as a promising class of semiconductor materials due to their versatility in applications ranging from field-effect transistors to optoelectronic devices [1,2,3,4]. The processability of CPs enables these materials to exhibit carrier mobility and optoelectronic properties that rival those of traditional inorganic semiconductors. Chemical doping is one of the critical enhancement techniques for CPs [5,6], which significantly increases their carrier concentration and conductivity, thereby improving their thermoelectric properties [7,8].

Unlike the atom substitution mechanism typical in inorganic semiconductors, doping in organic materials involves a reaction with the dopant, leading to intermolecular charge transfer [9,10]. This method, while effective in increasing carrier concentration, often disrupts the ordered stacking within the polymer, adversely affecting its transport properties due to complex non-covalent interactions between the dopant and the polymer [11,12]. Despite the existence of numerous ordered polymers that have undergone pre-treatment to achieve optimal structural integrity, including C_60_-based materials [13] and graphene-based materials [14], among others, there remains a paucity of comprehensive reports on this subject. Therefore, a detailed study of the impact of various dopants on the structural integrity of CPs is crucial for optimizing doping efficiency while minimizing structural damage [15,16].

Indacenodithiophene-*co*-benzothiadiazole (IDTBT) is a notable CP characterized by its novel charge transport mechanisms [17]. IDTBT maintains high carrier mobility even with lower crystallinity, attributed to its molecular architecture, which increases the rigidity of its aromatic groups and enhances interchain interactions [18]. These structural attributes contribute to high planarity and significant rotational torsion potential energy barriers, facilitating effective carrier transport despite its relatively low crystallinity. In this study, we examined the effects of two contrasting dopants, ferric chloride (FeCl_3_) and triethyloxonium hexachloroantimonate (OA), on the properties of the conjugated polymer IDTBT. Despite both dopants possessing potent oxidizing capabilities, they differ significantly in molecular structure and behavior. FeCl_3_ is a widely utilized high-efficiency dopant known for its robust performance in various organic polymers, while OA has been reported to enhance the planarity and electrical transport properties of the IDTBT polymer chain significantly [19]. Our research reveals that IDTBT doped with FeCl_3_ shows superior structural order and main-chain planarity, which correlate with increased carrier mobility and better intrinsic electrical transport properties. These insights are crucial for the strategic selection of dopants in the development of advanced conjugated polymers. The findings from this study not only deepen our understanding of dopant–polymer interactions but also guide future applications in organic electronics.

## 2. Materials and Methods

### 2.1. Materials’ Preparation

Indacenodithiophene-*co*-benzothiadiazole (IDTBT) was sourced from Derthon Optoelectronic Materials Science & Technology in Shenzhen, China, boasting a molecular weight over 50,000 and a polydispersity index (PDI) under 3. Iron(III) chloride (FeCl_3_) and triethyloxonium hexachloroantimonate (OA) were procured from Sigma-Aldrich. Solvents used included chlorobenzene (CB) for dissolving IDTBT and acetonitrile (ACN) for the dopants, also obtained from Sigma-Aldrich in Shanghai, China.

IDTBT solutions were prepared at 10 mg/mL in chlorobenzene and heated at 80 °C overnight before application. FeCl_3_ solutions were mixed to a 10 mM concentration and OA solutions to 10 mg/mL, both in acetonitrile. FeCl_3_ solutions were freshly prepared immediately before use to ensure stability. All preparative steps were conducted under a nitrogen atmosphere to maintain ultra-low moisture and oxygen levels (<1 ppm during solution preparation and <10 ppm during weighing).

To explore the impacts of FeCl_3_ and OA [20] on the structural and transport properties of IDTBT films, we meticulously prepared the samples through a controlled spin-coating process. The films were spun at 1500 rpm for 60 s to achieve uniform thickness. After spin-coating, the films were subjected to two different doping protocols using solutions of FeCl_3_ and OA dissolved in acetonitrile. The doping of the FeCl_3_ solution occurred over a range of 1 to 300 s, optimizing the interaction time for effective doping. Conversely, the OA solution required a substantially longer exposure time ranging from 0.5 to 36 h, tailored to its slower diffusion rate and interaction dynamics with the IDTBT polymer matrix. This sequential doping strategy, depicted in Figure 1a,b, was carefully designed to assess the distinct effects of each dopant on the film’s properties.

### 2.2. Testing and Characterization

#### 2.2.1. Spectroscopy Methods

X-ray photoelectron spectroscopy (XPS) spectra were collected using an Escalab 250xi in the Fujian Institute of Research on the Structure of Matter, Chinese Academy of Sciences. Parameters included a pass energy of 20 eV, a step size of 0.1 eV, and a spot size of 400 µm, with 30 scans per sample. Sulfur 2p spectra were analyzed for doublets with a 2:1 area ratio and spin–orbit coupling constant of 1.18 eV, applying reasonable constraints on line width variations. Data processing was carried out using ThermoAvantage software v5.9921.0.6648.

#### 2.2.2. Electrical Characterization

Electrical conductivities were determined using the equation σ = (R_s_ × t) − 1, where Rs is the sheet resistance, measured via the four-probe method, and t is the film thickness, measured using a DEKTAK XT step meter. The Seebeck coefficient measurements utilized a ZEM-3 device, equipped with two Peltier modules for temperature difference generation and two T-type Cu/Constantan thermocouples for simultaneous temperature z (Keithley 2400) and Seebeck voltage (Agilent 24700a) detection in the Fujian Mindu Innovation Laboratory, China.

#### 2.2.3. Structural Analysis

Raman spectroscopy was performed using an Alpha300R system from Witec with a 532 nm laser. Grazing-incidence wide-angle X-ray scattering (GIWAXS) was executed at the Fujian Mindu Innovation Laboratory, China, with an incident angle of 0.18° and an exposure time of 1200 s. The resulting 2D scattering images were analyzed using GIWAXS-tools software to fit line profiles to Gaussian functions for determining π-stacking peak widths and positions, facilitating the calculation of π–π stacking paracrystallinity using the formula
$$g=\frac{1}{2\pi }\sqrt{\Delta_{q}d_{\mathrm{hkl}}}$$
where $\Delta_{q}$ is the peak’s full width at half-maximum, and $d_{hkl}$ is the interplanar distance.

## 3. Results and Discussion

### 3.1. Charge Transport Analysis

The primary goal of this study was to systematically explore the effects of varying doping levels on the electrical properties of indacenodithiophene-*co*-benzothiadiazole (IDTBT) films. This was achieved by adjusting the concentration of the dopant solution and the duration of doping, utilizing two distinct dopants. The films were then characterized by measuring their Seebeck coefficient and conductivity at ambient temperature, with results analyzed using the charge transport model developed by Kang and Snyder et al. [22].

According to the Kang–Snyder model, the Seebeck coefficient and conductivity are inter-related through the transport parameter s and the transport coefficient σ_E0_. The parameter s reflects the dominant scattering mechanisms affecting carrier mobility, where s = 1 corresponds to metallic conduction dominated by acoustic phonon scattering, and s = 3 suggests thermally activated transport, typically influenced by ionized impurity scattering [23]. It is noteworthy that these relationships may not always align with the behavior of organic semiconductors due to their unique material properties [14].

In our study, σ_E0_, which is independent of energy but varies with temperature, serves as an indicator of intrinsic electrical transport properties. The data from the doped IDTBT films revealed significant discrepancies in the S–σ curves: films doped with FeCl_3_ aligned with s = 1, indicating a metallic conduction characteristic, whereas those doped with OA exhibited an s = 3 profile, indicative of thermally activated transport mechanisms (Figure 1c). These variations highlight the profound impact of dopant type on the charge transport properties of IDTBT.

To further understand these observations, additional S–σ data were collected from other researchers. Data from IDTBT mixed with OA, reported by Kim et al. [19], confirmed the s = 3 fit, aligning with our findings of thermally activated transport. Conversely, data from IDTBT sequentially doped with FeCl_3_, as described by Min et al. [21], consistently showed an s = 1 fit (Figure 1c), reinforcing the model’s validity and the influence of doping strategy on transport properties.

These findings underscore the complex interplay between dopant, doping method, and the resultant charge transport characteristics in organic semiconductors. They affirm the need for the precise control and understanding of doping conditions to tailor the properties of conjugated polymers for specific applications, furthering their potential for industrialization in optoelectronic devices.

Then, we primarily focused on the conductivity (σ) of indacenodithiophene-*co*-benzothiadiazole (IDTBT) films doped with different dopants and analyzed them across various temperature gradients from 155 K to room temperature. By examining the temperature dependence of σ, illustrated in Figure 2a,b, we ascertained the charge transport mechanisms within the doped IDTBT films [24,25]. The behavior typical of hopping transport, characterized by dσ/dT > 0, was observed across all doping levels for the three dopants used [26,27,28].

Further insights were gained by calculating the activation energy (Ea) from the slope of the Arrhenius plot (Figure 2c). This analysis helped determine the energy threshold required for carrier excitation and subsequent transport within the films, providing a comparative measure of structural ordering induced by different doping levels. The E_a_ values were inversely related to both the doping level and conductivity, reflecting the energy and permeability thresholds needed for carriers to transition to free carriers and facilitating transport within the permeation network [28].

Notably, the activation energy for the FeCl_3_-doped system exhibited lower sensitivity and consistently reduced values at low to medium doping levels compared to those of the OA-doped system. This suggests that the structural impact of FeCl_3_ doping on IDTBT is significantly less detrimental than that caused by OA, implying a gentler alteration in the polymer matrix.

Additionally, the W_γ_ values [25], indicative of the degree of energy disorder within the doped IDTBT, were computed and compared between doping systems. As presented in Figure 2d,e, W_γ_ provides qualitative insights into the energy disorder and the structural dynamics facilitating carrier transport between the crystalline and amorphous regions of the polymer [29]. Our results highlighted that OA doping resulted in higher W_γ_ values at both low and medium doping levels (Figure 2f), corroborating the observed activation energies and suggesting greater structural disruption and energy disorder in OA-doped films.

Overall, this detailed analysis underscores significant variances in how different dopants influence the electronic and structural properties of IDTBT, guiding future dopant selection to optimize charge transport and structural integrity in semiconductor applications [30,31].

### 3.2. Spectroscopy Analysis

The observed phenomena—increased energy disorder and reduced transport efficiency in IDTBT films—can largely be attributed to the polymer’s dependence on a highly planar backbone for effective charge transport. It is hypothesized that doping with OA at low and medium levels disrupts this planarity, leading to decreased structural order. This disruption hinders effective charge transport between the crystalline and amorphous domains within the polymer, significantly elevating the overall energy disorder.

To investigate these effects further, we employed Raman spectroscopy as a tool to quantitatively assess changes in the backbone planarity of the IDTBT molecule. This technique specifically measures the degree of torsion in the molecular backbone by analyzing two critical peaks in the Raman spectrum. The peaks, located at 1540 cm^−1^ and 1612 cm^−1^, correspond to the BT ring stretching mode and the IDT ring stretching mode, respectively [15,19,30,32,33]. The intensity ratio of these peaks (I_1540_/I_1612_) serves as an indicator of the torsion angle between the IDT and BT rings, with a higher ratio suggesting less torsion and greater planarity of the main chain [32,33].

This method allowed us to directly correlate structural changes within the IDTBT polymer to the doping levels applied, providing a clearer understanding of how doping affects the molecular structure and, consequently, the transport properties of the material. The outcome of this analysis would confirm or refute our initial speculations regarding the impact of OA doping on the planarity of the IDTBT backbone and its subsequent influence on the polymer’s electronic properties.

Our investigation included a comprehensive analysis of multiple IDTBT samples, each representing different conductivity levels from two distinct doping systems (Figure 3a,b). The data revealed a pattern where the planarity of IDTBT’s main chain initially increased with rising doping levels but subsequently decreased (Figure 3c). This pattern aligns with findings from prior research suggesting that IDTBT inherently possesses a compact and rigid main-chain structure. However, its branched chains, which are long and flexible, typically adopt a less ordered arrangement [34]. During doping, counterbalance ions introduced by the dopant preferentially infiltrate these branched regions, which have larger free volumes. As doping progresses, the occupancy of these regions by counterbalance ions supports the main chain structurally, enhancing torsional rigidity and potentially increasing the density of cross-links, thus providing stable paths for interchain transport. However, if this equilibrium is destroyed by the increase in counterbalance ions, the result is instead a reduction in the planarity of the main chain due to excessive torsional stress and gradually occupied main-chain regions [35,36].

Further investigations focused on the impact of doping on the main-chain planarity, particularly comparing IDTBT doped with FeCl_3_ and OA. Raman spectroscopy analyses showed that peak-to-intensity ratios were consistently lower in the OA-doped samples compared to those in samples doped with FeCl_3_, indicating a less adverse impact on main-chain structure by FeCl_3_ (Figure 3c). To explore whether discrepancies in main-chain planarity underlie variations in mobility, we selected two samples with identical conductivities for deeper analysis. The Raman spectra confirmed that both doping systems achieved optimal main-chain planarity at these conductivity levels.

Subsequent X-ray photoelectron spectroscopy (XPS) analysis focused on approximating the carrier concentration and mobility by examining the S2p peaks [37,38,39] (Figure 3d,e). The results demonstrated that at the same level of conductivity, IDTBT samples doped with FeCl_3_ exhibited superior carrier concentration and mobility compared to those doped with OA (Figure 3f). This implies that FeCl_3_ not only enhances doping efficiency but also promotes a more ordered structural arrangement within the polymer.

To characterize the electron transfer and charge carrier generation induced by FeCl_3_ and OA dopants in IDTBT films, we extended our analysis to include UV–vis–NIR absorption spectra (Figure 3g). The spectra revealed a notable bleaching at the 650 nm peak in the FeCl_3_-doped films, indicative of significant oxidation and radical cation formation. This oxidation led to a decrease in the HOMO energy level, enhancing hole mobility. The observed spectral features, such as the optical transitions designated as P1 and P2, were more pronounced in the FeCl_3_-doped films. P1 transitions, seen at about 1300 nm, and P2 transitions, observed at 1060 nm, signify shifts in polariton bands and are attributed to the elevated levels of radical cations. These results underscore the critical role of dopant-induced electronic structure modification in enhancing the conductivity and overall electronic properties of polymer films.

In detail, FeCl_3_ serves as an oxidizing agent that initiates a reversible electron transfer reaction. This reaction leads to the generation of positively charged radical cations within the polymer. Subsequently, the FeCl_4_^−^ ion, produced during the reaction, enters the polymer matrix as an ionized dopant. This ion compensates for the positive charges formed on the polymer backbone, maintaining electrical neutrality. Additionally, the doping process facilitates the formation of polariton states within the intrinsic bandgap of the polymer. These polaritons are quasiparticles resulting from strong coupling between the electronic excitations of the polymer and photons. The formation of these polariton states not only alters the optical properties of the material but also necessitates energy expenditure to accommodate the structural distortions caused by the doping.

The differences in transport coefficients observed between the two doping systems primarily stem from their distinct impacts on the main-chain planarity of IDTBT molecules. As evidenced by Raman spectral analysis and corroborated by the data in Figure 1c, IDTBT, a polymer that critically depends on planar main chains for effective carrier transport [40] exhibits significantly altered transport properties when doped with different agents. The OA doping system, in particular, leads to a noticeable deterioration in the planarity of the main chains compared to FeCl_3_ doping. This reduction in planarity is a key factor contributing to the observed decline in the transport σ_E0_.

This detailed examination of the molecular structure through Raman spectroscopy provided crucial insights into how doping affects the structural integrity of polymer chains. The results clearly demonstrate that the inferior main chain planarity resulting from OA doping undermines the intrinsic electrical transport properties of IDTBT, highlighting the importance of selecting dopants that preserve or enhance the molecular structure to optimize electronic performance.

### 3.3. GIWAXS Analysis

To further investigate the impact of main-chain planarity on charge transport, we employed grazing incidence wide-angle X-ray scattering (GIWAXS) to characterize the microstructure of the IDTBT films. Typically, IDTBT films exhibit poor crystallinity, with GIWAXS images showing no substantial differences between the crystalline and amorphous regions in undoped samples [41]. Notably, the (010) peak, indicative of π–π stacking in the out-of-plane direction, appears broad and exhibits low intensity, suggesting diffuse aggregation (Figure 4a). Conversely, the (00I) peak, associated with main-chain stacking in the in-plane direction, is more pronounced, demonstrating stronger peak intensity.

Upon doping, significant differences emerged between films treated with FeCl_3_ and those treated with OA [42]. Specifically, OA-doped IDTBT films displayed a (200) peak that signified lamellar structure in intermolecular stacking, suggesting a more ordered arrangement compared to the undoped state (Figure 4b,c). Moreover, the d spacings of the (100) and (010) peaks in OA-doped films were found to be larger [19], indicating expanded molecular separation, and the overall crystallinity was reduced (Figure 4e). The disruptive impact of OA’s larger molecular size when incorporated into polymer films is evident, leading to structural alterations, as observed in our and other studies [8]. However, it is critical to note that, at lower doping levels in IDTBT, the relationship between dopant size and polymer crystallinity—while significant—is not the only factor influencing structural integrity [37]. Our findings suggest that the molecular size of OA contributes to, but does not solely dictate, the differences in structural damage when compared with smaller dopants like FeCl_3_. This nuanced understanding is supported by evidence of reduced crystallinity and interchain order, as demonstrated in the GIWAXS and corroborated by Raman spectroscopy results, which revealed decreased backbone planarity and disrupted intermolecular stacking. These structural changes are pivotal in understanding dopant–polymer interactions and their implications for electronic properties. This manuscript has been enhanced with additional references and detailed explanations to clarify these complex phenomena, making it accessible and informative for readers seeking to understand the multifaceted impacts of dopant size on polymer systems. In summary, this expansion and reduction in crystallinity contribute to substantial disruptions in interchain transport, aligning with the findings from Raman spectroscopy, which indicated diminished backbone flatness and disrupted intermolecular stacking.

The GIWAXS data further revealed that OA doping led to a marked decrease in overall peak intensity, implying a significant disruption in the structural order and potentially partial disintegration of the polymer chain structure. This was corroborated by a pronounced reduction in the crystallinity of the in-plane (001) peak and an expansion in the d spacing of the (003) peak, indicating weakened stacking of the primary chains following OA doping [21] (Figure 4d).

In conclusion, these microstructural changes suggest that OA-doped IDTBT is not conducive to effective one-dimensional charge transport due to its lower main-chain planarity and disrupted molecular stacking compared to FeCl_3_ doping. These findings emphasize the critical role of dopant selection in preserving or enhancing the structural integrity necessary for optimized electronic transport in conjugated polymers.

## 4. Conclusions

This study highlights the significant variability in the impact of different dopants on the intrinsic electrical transport properties of conjugated polymers (CPs), even when applied at equivalent doping levels. This variability largely stems from the structural characteristics of the polymers, particularly the planarity and rigidity of the main chain.

IDTBT, a CP characterized by its highly planar main chain and substantial rotational and torsional potential energy, serves as a critical conduit for one-dimensional charge transport. The planarity of the main chain crucially influences IDTBT’s intrinsic electrical properties, while its rigidity helps to buffer the structural perturbations induced by chemical doping. Although excessive doping can compromise the planarity of the IDTBT backbone, the selection of appropriate dopants can preserve backbone rigidity, thus enhancing transport properties and even enabling metallic conduction characteristics.

In this study, we meticulously analyzed the impact of FeCl_3_ and OA doping on the thermal and electrical properties of IDTBT films. By integrating temperature-dependent mobility assessments and molecular structural analyses through Raman spectroscopy and grazing incidence wide-angle X-ray scattering (GIWAXS), we established distinct advantages in the electrical transport properties associated with FeCl_3_ doping. Specifically, FeCl_3_-doped IDTBT films achieved a conductivity of up to 16.1 S/cm—marking the highest recorded for such films—and demonstrated enhanced main-chain planarity. Conversely, OA-doped films, while beneficial, showed a lesser degree of these enhancements. These findings highlight the critical role of dopant selection in optimizing the transport properties of organic semiconductors, potentially elevating them to levels observed in inorganic semiconductors. Our research illustrates that with strategic dopant selection, IDTBT films can achieve transport properties with a semi-metallic signature, signifying a significant leap in the capabilities of organic semiconductor materials. This conclusion not only underscores the transformative potential of precise dopant application but also lays the groundwork for future investigations into the optimization of thermal and electrical properties in organic semiconductors. In summary, our research demonstrates that judiciously chosen dopants can elevate the transport properties of organic semiconductors to levels comparable to those of inorganic semiconductors. This finding suggests that, with optimal dopant selection, IDTBT films could potentially achieve “electron crystal” transport properties, a notable advancement in the field of organic semiconductors.

Moreover, our exploration into the doping strategies and molecular structures of CPs suggests that, in the near future, conductive CPs could surpass the thermoelectric performance of commercially available PEDOT-based derivatives in thermoelectric applications. This prospective leap in performance underscores the transformative potential of targeted chemical doping in enhancing the functional capabilities of CP-based technologies.

## Figures and Tables

**Figure 1 polymers-16-02467-f001:**
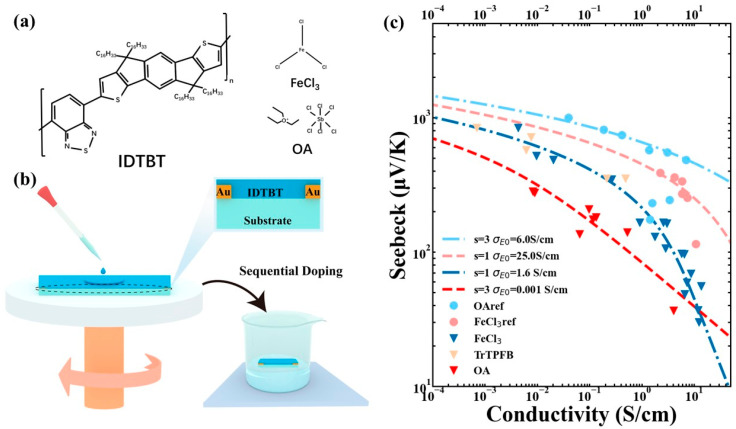
(**a**) Chemical structures of host and dopants used in this study; (**b**) a schematic diagram of the spin-coated film, sample preparation, and doping methods used in this experiment. (**c**) Experimental (symbols) and calculated (dotted lines) relation of Seebeck coefficient S versus electrical conductivity σ including compilation of literature IDTBT data and our data. Parameters (s, σ_E0_) used for the Kang–Snyder model fitting are given in the figure; FeCl_3_-doped (blue triangles, our data), OA-doped (red triangles, our data), FeCl_3_-doped (blue circles, ref. [21]), OA-doped (red circles, ref. [19]).

**Figure 2 polymers-16-02467-f002:**
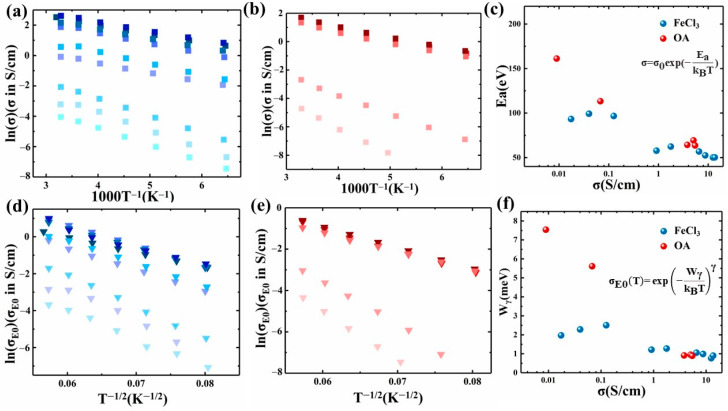
Charge transport mechanisms of chemically doped IDTBT. (**a**,**b**) Temperature (T) dependence of electrical conductivity σ and (**c**) activation energy E_a_ extracted from Arrhenius plot of electrical conductivity σ. (**d**,**e**) Temperature (T) dependence of transport coefficient σ_E0_(T) and (**f**) activation energy W_γ_ extracted from the temperature dependence of transport coefficient σ_E0_(T) with γ = 1/2.

**Figure 3 polymers-16-02467-f003:**
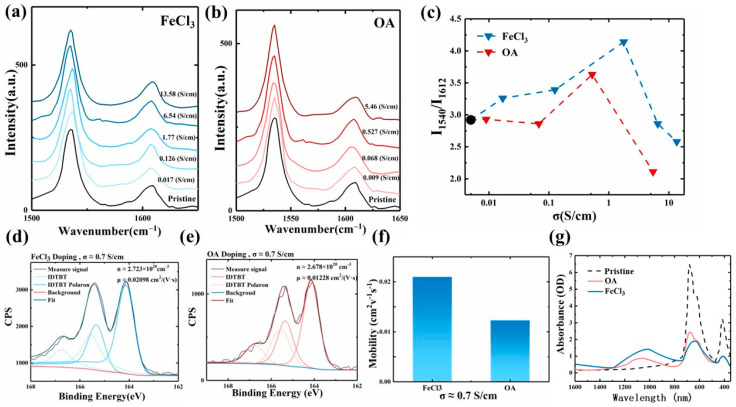
(**a**,**b**) Raman spectra of undoped and doped IDTBT film. (**c**) Relative intensity ratio (I_1540_/I_1612_) of Raman peaks at 1540 cm^−1^ (I_1540_) and 1612 cm^−1^ (I_1612_), calculated from the Raman spectra in (**a**,**b**). Carrier density measurement in IDTBT. Sulfur 2p XPS spectra of doped IDTBT films: (**d**) FeCl_3_ doping; (**e**) OA doping; (**f**) mobility calculated from the XPS; (**g**) UV–vis–NIR absorption spectra.

**Figure 4 polymers-16-02467-f004:**
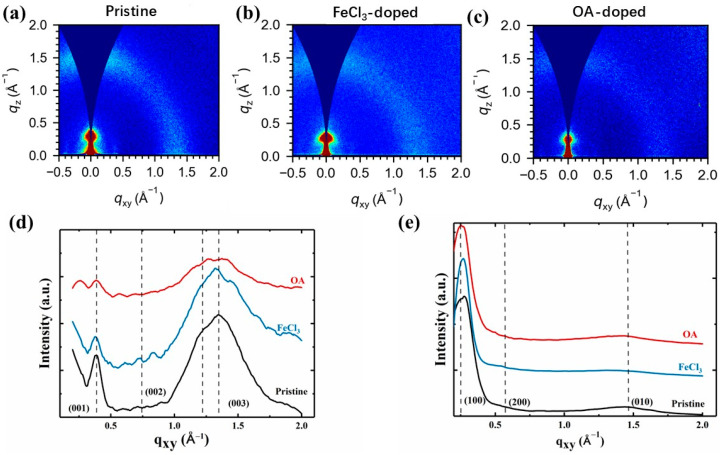
GIWAXS patterns of (**a**) undoped and (**b**,**c**) doped IDTBT films. GIWAXS line-cuts of (**d**) in-plane and (**e**) out-of-plane directions of undoped and doped IDTBT films. Data are the same as used in (**a**–**c**).

## Data Availability

Data are contained within this article.

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
