# Peer review of "Impact of Structural Alterations from Chemical Doping on the Electrical Transport Properties of Conjugated Polymers"

_polymers, 2024, doi:10.3390/polym16172467_

Round 1

Reviewer 1 Report

Comments and Suggestions for Authors

Lin and co-workers report the impact of structural alterations from doping on the electrical transport properties of conjugated polymers. The manuscript needs improvement on the following lines.

1. The reaction of the dopant with the donor organic polymer has not been discussed. In general, organic donors with suitable HOMO/valence bands undergo oxidation in presence of FeCl3, Cu(II), etc. forming radical cations promoting hole formation and its transport. This aspect must be discussed and clarified in the results and discussion. The authors should try to record EPR spectrum, of the oxidized thin films.

2. UV-Vis-NIR absorption spectrum should be recorded of the doped thin films as characteristic radical formation or mixed valence states can be determined as well-known for organic donors (refer to above ref.). 

3. What is the nature of electronic interaction authors envisage between the product of FeCl3 oxidation, i.e., FeCl4- ? and the polymer. This should be discussed.

4. Some of the well-known organized polymers made from semiconductive C60 see ref. J. Mater. Chem. 2007, 17, 2454, should be referred to in the introduction.

Comments on the Quality of English Language

Minor improvements can further improve the manuscript.

Reviewer 2 Report

Comments and Suggestions for Authors

In the manuscript entitled "Impact of Structural Alterations from Chemical Doping on the Electrical Transport Properties of Conjugated Polymers", the authors investigated the microstructural changes upon p-doping of IDTBT with two different inorganic dopants: FeCl3 and trimethylindium hexachloroantimonate (OA). They evaluated the effectiveness of doping as a function of conductivity and Seebeck coefficient, and correlated the conductivity data of the two dopants with the packing structure of the IDTBT polymer film using advanced techniques such as GIWAXS. I found the research to be thorough and of interest to the conjugated polymer community. However, there are several issues, as listed below, that need to be addressed before recommending it for publication.

1. In the introduction, the authors should also explain why the two particular dopants were chosen. Also, at the end of the introduction, please include a paragraph stating the main findings/results of the research.

2. It has been found that at the same doping level, OA causes greater microstructural changes such as increased d-space, reduced crystallinity. Is this because OA has a much larger molecular size than FeCl3? Is there any literature on this?

3. In session 2.1 Materials Preparation, a representative procedure with specific parameters of spin coating to prepare the IDTBT film and then the sequential doping procedure with FeCl3 and OA, respectively, should be given.

4. In Figure 1c, the axis tickers are not well visible.

5. Figure 4b, FeCl3

6. The conclusion is written in very general terms. This is not wrong per se, but one should also add specific findings/results of the research, especially if it is a research article. The two dopants used and the results (conductivity, Seebeck coefficient) are completely missing in the conclusion.
